# Multifaceted Nature of Lipid Droplets in Viral Interactions and Pathogenesis

**DOI:** 10.3390/microorganisms11071851

**Published:** 2023-07-21

**Authors:** Luis Herrera-Moro Huitron, Luis Adrián De Jesús-González, Macario Martínez-Castillo, José Manuel Ulloa-Aguilar, Carlos Cabello-Gutierrez, Cecilia Helguera-Repetto, Julio Garcia-Cordero, Moisés León Juárez

**Affiliations:** 1Laboratorio de Virología Perinatal y Diseño Molecular de Antígenos y Biomarcadores, Departamento de Inmunobioquímica, Instituto Nacional de Perinatología, Mexico City 11000, Mexico; luis_5m5@hotmail.com (L.H.-M.H.); josemanuel7111@gmail.com (J.M.U.-A.); 2Unidad de Investigación Biomédica de Zacatecas, Instituto Mexicano del Seguro Social, Zacatecas 98000, Mexico; adrian_6101@hotmail.com; 3Sección de Estudios de Posgrado e Investigación, Escuela Superior de Medicina, Instituto Politécnico Nacional, Mexico City 11340, Mexico; 4Instituto Nacional de Enfermedades Respiratorias Ismael Cosío Villegas (INER), Departamento de Investigación en Virología y Micología, Calzada de Tlalpan 4502, Belisario Domínguez, Tlalpan 14080, Mexico; carloscginer@gmail.com; 5Laboratorio de Microbiología y Diagnóstico Molecular, Departamento de Inmunobioquímica, Instituto Nacional de Perinatología, Mexico City 11000, Mexico; ceciliahelguera@yahoo.com.mx; 6Departamento de Biomedicina Molecular, Cinvestav, Av. IPN# 2508, Mexico City 07360, Mexico; leebeydengue@yahoo.com.mx

**Keywords:** lipid droplets, viral infections, pathogenesis, immune response

## Abstract

Once regarded as inert organelles with limited and ill-defined roles, lipid droplets (LDs) have emerged as dynamic entities with multifaceted functions within the cell. Recent research has illuminated their pivotal role as primary energy reservoirs in the form of lipids, capable of being metabolized to meet cellular energy demands. Their high dynamism is underscored by their ability to interact with numerous cellular organelles, notably the endoplasmic reticulum (the site of LD genesis) and mitochondria, which utilize small LDs for energy production. Beyond their contribution to cellular bioenergetics, LDs have been associated with viral infections. Evidence suggests that viruses can co-opt LDs to facilitate their infection cycle. Furthermore, recent discoveries highlight the role of LDs in modulating the host’s immune response. Observations of altered LD levels during viral infections suggest their involvement in disease pathophysiology, potentially through production of proinflammatory mediators using LD lipids as precursors. This review explores these intriguing aspects of LDs, shedding light on their multifaceted nature and implications in viral interactions and disease development.

## 1. Introduction

### 1.1. The Structure of the Lipid Droplets Makes It a Unique and Special Organelle

Lipid droplets (LDs) are distinctive organelles within eukaryotic cells and are identified by their unique structure. Unlike most cellular organelles, which typically possess a lipid bilayer, LDs have a characteristic lipid monolayer and a core primarily composed of neutral lipids, such as triacylglycerols and cholesterol esters. They also contain other lipids, such as polyunsaturated fatty acids, one of the most abundant classes of eicosanoids [1].

The quantity and variety of lipids stored in LDs depend on two key factors: the cell lineage or tissue and the energy demand of the cells. Another defining characteristic of these cellular organelles is the presence of various proteins within the lipid monolayer. These proteins are vital to numerous cellular functions, ranging from those involved in lipid signaling and metabolism, gene transcription, and autophagy for cells contributing to membrane trafficking and innate immunity. Approximately 150 proteins have been reported to be harbored within LDs [2,3].

Moreover, LDs are highly hydrophobic structures with an oily phase and an aqueous phase within their constitution. This dual-phase nature facilitates their interaction with other cellular organelles, leading to their characterization as dynamic organelles. One of the major organelles they interact with is the endoplasmic reticulum, where their biogenesis commences [4].

### 1.2. Endoplasmic Reticulum as a Precursor for Lipid Droplet Biogenesis

The biogenesis of LDs is significantly involved with the endoplasmic reticulum (ER), with enzymes integral to LDs biogenesis identified as anchored to the ER membrane [5]

The assembly of LDs involves a series of four well-established steps: (Figure 1).

Step one: Esterification of Fatty Acids

The initial step involves the esterification of fatty acids within the bilayer lamellae of the ER, facilitated by enzymes involved in lipid biogenesis, such as Acetyl-Coenzyme A acetyltransferase (ACAT) and diacylglycerol acyltransferases (DGAT). Two isoforms of DGAT (DGAT1 and DGAT2) have been identified and are known to play active roles in the biogenesis of LDs. DGAT1 is anchored to the ER membrane as either a dimer or tetramer. Its primary substrate is acyl-CoA, which it processes to convert diacyl-glycerols into triacylglycerols, the principal monomers of LDs [6,7,8,9].

A unique trait of DGAT1 is its ability to aid in LD formation and protect cells against the detrimental effects of lipid accumulation. In contrast, DGAT2 is positioned at two different sites. It resides in the ER, where it transforms diacylglycerols into triacylglycerols and is also located within the lipid droplets themselves, promoting their growth. Another crucial element in this process is the formation of cholesterol esters within the ER, facilitated by the enzyme cholesterol acetyltransferase. This enzyme catalyzes the conversion of two units of acetyl-CoA into acetoacetyl-CoA, a critical step in the mevalonate pathway, an essential metabolic route found in all eukaryotes [10].

Step Two: Formation of Lens-Like Structure

In the second step, the triacylglycerols and cholesterol esters formed in the ER lamellae coalesce, forming a lens-like structure. This process is facilitated by the curvatures and morphology of the ER, initiated by microtubules, which give rise to a lipid aggregate of approximately 30 nm. It has been reported that there is a threshold to the number of triacylglycerols that can aggregate before the lens-like emulsion separates from the inner membranes of the ER, with the limit being approximately 3% of the total triacylglycerols present in the ER’s inner membranes [11,12,13].

As the ER extends throughout the cell, it provides a conducive environment for accumulating triacylglycerols. These aggregates depend on the microtubule network present in the ER, which is maintained by proteins of the reticulon family and atlastins [14]. The network’s importance in LD formation is so significant that, in the event of ER microtubule degradation, specific proteins, such as receptor accessory protein type 1 (REEP1), promote microtubule regeneration [15].

A pivotal step in forming the lipid lens is the interaction of the fat storage-inducing transmembrane protein 2 (FIT2). Anchored to the ER membrane, FIT2 can associate with diacylglycerols, favoring the binding of DGAT enzymes, thus promoting the progression of LD formation [16].

Step Three: Budding of New LDs

The third step involves the budding of new LDs, a process triggered when sufficient lipid accumulation is achieved within the lens-like structures formed in the ER bilayer lamellae [13,17]. This accumulation creates surface tension on the ER lamellae, further amplified by phospholipids, which assist in the budding of nascent LDs [18,19,20].

A noteworthy characteristic of LDs in eukaryotic cells is their complete budding, unlike yeast, where new lipid droplets remain partially anchored in the ER. This full budding is facilitated by transmembrane proteins in the ER, such as FIT2 proteins. FIT2 proteins promote budding and influence where an LD can originate, as LD formation is restricted to specific ER sites where these anchored proteins are found [21].

In this context, Seipin, a protein with transmembrane domains, is pivotal in facilitating the budding and directional formation of de novo LDs. Seipin is a highly conserved protein across eukaryotic and prokaryotic cells, present in all cell lineages. However, the quantity of Seipin varies among cell types; for example, it is abundant in neurons while less prevalent in adipocytes [22,23]. Studies reveal that a deficiency in Seipin leads to an accumulation of small LDs in the ER, promoting cell lipotoxicity. These findings underscore the significance of proteins such as seipin in not only assisting the generation of nascent LDs but also facilitating their growth and expansion [13].

Upon release into the cytoplasm, nascent LDs can immediately interact with specific proteins. One such protein is glycerol-3-phosphate acyltransferase 4 (GPAT4), which facilitates the recruitment of triacylglycerols [24,25].

Step Four: Expansion and Growth of LDs

The fourth and final step in LD formation involves expansion and growth. Newly formed LDs can interact and transfer triacylglycerols to each other within just a few minutes of their formation, a phenomenon observed in mammalian and insect cells [12]. In addition, the outer membrane of the ER provides phospholipids to nascent LDs, supporting their growth and expansion [26].

Once formed, LDs can be categorized into two distinct subpopulations. The first comprises initial lipid droplets (iLDs) that bud from the ER, and the second is expanding lipid droplets (eLDs) that grow and expand with resources supplied by the ER and the cytoplasm [27].

In this context, two classes of LD-associated proteins have been identified. Class I proteins, such as DGAT2, ATGL, UBXD8, and ACSL3, originate from or are supplied by the ER. These proteins foster the growth and expansion of LDs and facilitate interaction with some regions of the ER. Furthermore, they can bind to LDs through hydrophobic bonds, positioning themselves close to the LD core [21].

Class II proteins associated with LDs originate from the cell cytoplasm. The Perilipin family proteins (PAT) are prominent among these, which shield LDs from lipase activity, including ATGL and HSL. These class II proteins also enhance interaction with other cellular organelles and participate in the LD degradation process known as lipophagy. Like class I proteins, they also bind to LDs through hydrophobic bonds. However, it is worth noting that the hydrophobic domains facilitating the binding of class II proteins are longer than those in class I. Proteomic studies have shown that between 100 and 150 proteins are associated with LDs in eukaryotic cells. This association grants them a higher level of dynamism among themselves and most organelles, compared to within prokaryotic cells, where only 35–40 proteins are associated with LDs [14].

Moreover, the formation of giant LDs has been observed in specific cell lineages, such as adipocytes and hepatocytes. LD size can reach up to 2 μm in these cells, achieved through the coalescence of two large LDs. Another mechanism for this involves a single LD attracting smaller LDs until they encompass a giant LD [21].

## 2. Exploring the Multifaceted Roles of Lipid Droplets and Their Interactions with Cellular Organelles

### 2.1. Multidimensional Role of LDs in Cellular Metabolism

In cellular metabolism, the LDs are the storage depots for neutral lipids. These lipids are mobilized and degraded through two major pathways: lipolysis and lipophagy. The lipolysis pathway primarily involves enzymes such as adipose triglyceride lipase (ATGL) and hormone-sensitive lipase (HSL). Lipophagy involves the formation of an autophagosome and the recruitment of autophagy-associated elements. Both pathways facilitate the release of fatty acids, which can act as intermediates in the beta-oxidation process, providing energy when needed by the cell [28,29,30].

Beyond their role in energy metabolism, LDs are instrumental in maintaining lipid homeostasis. They influence lipogenesis and lipolysis within cellular metabolism, crucial in safeguarding the organism from diseases such as diabetes, heart disease, and particularly obesity [31,32]. Additionally, by retaining lipids such as phospholipids, LDs prevent degradation and oxidation, ensuring they can be repurposed for membrane formation [33,34,35].

Recent research extends the functional repertoire of LDs beyond lipid homeostasis in mammalian cells, highlighting their role as a detoxification system in the symbiosis between fungi and lichens. For example, LDs have been observed to mitigate the oxidative stress caused by perylene-quinones (toxins) released by the *Phaeosphaeria* genus of fungi, which produce reactive oxygen species (ROS). LDs can sequester these toxins, reducing ROS levels and preserving the symbiotic relationship [36].

Furthermore, emerging evidence suggests that LDs can modulate apoptosis. This has been exemplified in hepatocellular choriocarcinoma cells deficient in the PSMD1 and PSMD2 subunit genes participating in LD lipid metabolism. A decrease in LDs in these cells leads to the initiation of the apoptosis mechanism, as indicated by the increased expression of caspases. However, overexpression of these genes reverses this trend, restoring LD levels and reducing the expression of proteins linked with apoptosis [37].

### 2.2. Role and Participation of LDs in Immune System

Some of the extra functions of LDs unrelated to participating in the mobilization of energy reserves are regulating vesicular traffic, folding and storing proteins, and autophagy. However, in recent years, it has been found that LDs of immune cells, such as neutrophils and macrophages, have a role in activating and regulating the immune response to fight viral and bacterial infections (Figure 2) [38].

In that sense, a well-studied protein related to LDs and the activation of immunity is the protein viperin. Initially, viperin is synthesized after activation of TLR7 and TRL8 receptors, which can detect pathogens, after activation the protein is anchored to the ER and subsequently transported to the LDs, where it can interact with the kinase associated with the interleukin-1 receptor (IRAK) 1 and the TNF receptor-associated factor (TRAF6). Additionally, it is known that viperin can bind to STING to increase TBK1 ubiquitination and thus enhance the IFN type 1 response [39,40]. Another mechanism by which viperin acts is by interacting with the peroxisome protein Pex19. Pex19 is involved in forming early peroxisomes, which, like viperin, favor the innate immune response by activating the MAVS signaling pathway. It has been observed that LDs that are in association with viperin can translocate to peroxisomes if this complex is not formed. It is difficult to find a peroxisome associated with LDs; this fact shows the importance of LDs in activating ISGs against viral infections [41,42,43]. While viperin can bind to other proteins to trigger an interferon response, it has been observed that it can only translocate to LDs to catalyze CTP conversion to ddhCTP, which is a ribonucleotide that serves as a terminator of viral RNA synthesis [44]. However, viperin is not the only protein with antiviral properties that can be found in LDs, another protein with antiviral properties that have been found in LDs is cathelicidin protein (CAMP) that belongs to a group of antimicrobial peptides that can inhibit the viral and bacterial infection. The antimicrobial mechanism of CAMP is that it interacts with the cell membranes of microorganisms, causing damage/perforation. On the other hand, it has also been found that a specific group of histones can translocate to LDs [45,46]. However, it has not been shown that histones have antiviral activity. It has been found that histones H2 H2B, H3, and H4 can inhibit the replication of cells infected with IAV [47,48]. With this evidence, it is believed that histones may exist within LDs and may have an antiviral effect, but more research is needed to elucidate the role that histones may have during infection in LDs.

Although there are currently no more reports of proteins with antiviral properties residing or localizing in LDs during infections, there are studies that have demonstrated interaction with proteins that are already known to be resident and cover LDs, as is the case of the PLIN-2 protein. In this sense, it has been reported that cells are stimulated with LPS, TNF-α, and interferon-gamma; some proteins such as GVIN, IFGGA1, IFGGB55, and IFI35 are associated with PLIN2. In addition to finding IGTP, IIGP1, TGTP1, and IFI47 were associated with LD, suggesting a connection between LD and the immune system [49]. On the other hand, Monson and collaborators found that in cells infected with the ZIKV virus and Herpes simplex 1, there was an increase in LDs, and interestingly, in pharmacological inhibition of LDs, there was an increase in viral replication [50]. However, although there is invited evidence that LDs participate in immunological processes, the mechanism by which LDs regulate these processes remains unknown. Still, one of the hypotheses is that the induction of LDs can improve the interferon response because Monson and collaborators find an increase in interferon after stimulating the formation of LDs. In contrast, lowering LDs alters the regulation of ISGs, delaying the host’s antiviral response [51]. On the other hand, Bougneres and collaborators propose that dendritic cells with many LDs are more effective at presenting antigens to CD8 T cells because when the inhibition of the formation of LDs was performed, the antigen presentation showed an inhibitory effect [52].

Finally, other mechanisms by which LDs could exert a role as immunity mediators are eicosanoids, molecules derived from arachidonic acid (AA), and anti-inflammatory lipids, which can be derived from polyunsaturated fatty acids (PUFA). These molecules regulate the immunity associated with LDs because they can be obtained from LDs when they are degraded, so LDs would control the storage and release of inflammatory or anti-inflammatory precursors. However, more analysis is still needed to reach more specific mechanisms [53,54].

### 2.3. Interactions of LDs with Other Cells’ Organelles

LDs are also highly dynamic units within the cell, capable of interacting with most cellular organelles. LDs interact significantly with the ER, cytoskeleton, mitochondria, nucleus, peroxisomes, and lysosomes (Figure 3).

In the cytoskeleton, the motility of LDs is facilitated by a specialized system that relies on the cell’s cytoskeleton, particularly the microtubules. These microtubules serve as ways for LDs to travel to various cell parts. This movement is driven by motor proteins, primarily Dynein and Kinesin I, which associate with LDs and propel them toward other organelles [55]. While class II and V myosins have been implicated in LD motility, this phenomenon has been primarily observed in yeast [56].

Understanding the movement and interaction of LDs with other organelles provides essential insights into their role in cellular function and metabolism. The central organelle with which LDs interact is the ER since this is where they originate. However, LDs also establish crucial interactions with the mitochondria, a site of high energy consumption within the cell. This interaction allows for a necessary flow of fatty acids and lipids, providing substrates for energy generation via beta-oxidation. In turn, this helps to meet the cell’s energy demands. Some mitochondrial enzymes have been identified to interact with LDs, facilitating a bidirectional connection. Furthermore, proteins known as Perilipins have been reported to mediate the interaction between mitochondria and LDs. Among these, PLIN-5 plays a crucial role in maintaining the connection between these organelles [57,58].

LDs have also been found to associate with the cell nucleus. This connection is unexpected, given the direct link between the nucleus and the ER. Various analyses show that LDs exist both on the nuclear surface and within the nucleus. One of their primary functions within the nucleus is their interaction with specific transcription factors, aiding in the storage and proper functioning of histones vital for chromatin remodeling. CCT1, a rate-limiting enzyme in phosphatidylcholine biosynthesis, is another protein reported to travel from LDs to the nucleus [59,60].

In addition, some proteins have been found to colocalize with both the nucleus and LDs. An example is Prp19, a protein involved in the maturation of messenger RNA. This suggests that intricate interactions and molecular communication between LDs and the nucleus may occur in various directions and under different stimuli. Further research is required to elucidate these interactions and their implications [60].

Lipid droplets also engage in a critical interaction with peroxisomes, which organelles involved in various metabolic processes including β-oxidation. In yeast, peroxisomes assist in β-oxidation, while in mammalian cells, they are involved in the β-oxidation of long-chain and branched fatty acids. Several studies have identified contact sites between LDs and peroxisomes, suggesting that this interaction may facilitate enzyme exchange and thus regulate the degradation of fatty acids stored in LDs [61,62].

Finally, lysosomes maintain a close relationship with LD since, as mentioned above, one of the main functions of LD is to provide energy to the cell when it is required. The relevant mechanism involves lipophagy, which is known as selective autophagy in this process, in which the participating lysosomes can degrade LD through the formation of an autophagolysosome. However, it has been shown that LD-associated proteins such as perilipins, specifically perilipin 2 or ADFP, can mediate chaperone-mediated autophagy, where signaling is activated to initiate the process of lysosome-mediated lipophagy. It is a reality that LDs maintain a close relationship with the other cellular organelles. Their primary function is to provide lipids to keep the membranes properly functioning or to provide energy and serve as a signaling pathway. In conclusion, LDs uphold a close relationship with other cellular organelles [63,64].

## 3. Role of Lipid Droplets in the Viral Infection Cycle and Human Disease Progression

Advances in virology have revealed a series of strategies that viruses use to recruit cellular components to facilitate their replication and ensure a successful infection cycle. Among these components, lipids play a crucial role. LDs have been implicated in vital activities such as viral morphogenesis and the forming and releasing of viral particles. In addition, the breakdown of these lipid stores can be exploited by the virus as an alternative energy source. The following sections discuss several examples of how viruses that infect humans interact with LDs during infection (Figure 4).

### 3.1. Hepatitis C Virus (HCV)

Approximately 185 million people worldwide are infected with the hepatitis C virus (HCV), which can lead to severe conditions such as cirrhosis, fibrosis, nonalcoholic fatty liver disease (NAFLD), and liver inflammation [65]. Since the liver is responsible for regulating lipid homeostasis, HCV infection can affect this function and lead to an imbalance in lipid management, known as hepatic steatosis [66,67]. It has been observed that patients diagnosed with the infection exhibit increased lipid levels (approximately 40%). In addition, LDs were also increased, compared to other liver diseases [68].

Initial evidence of the interaction between lipids and HCV was revealed by Morapdour in 1996, who suggested that the virus core protein anchors to the surface of LDs during viral infection [69]. This finding was later supported by several studies indicating that the core protein, once cleaved, can associate with LDs via its two amphipathic helices and hydrophobic regions. Furthermore, it has been discovered that not only does the core protein interact with LDs but the NS5A protein does as well. This protein is composed of three domains, with the N-terminus responsible for the interaction with LDs [70].

This interaction between viral proteins and LDs favors the assembly of viral particles [69,70,71,72,73]. In addition, it has been shown that viral proteins, such as core and NS5A, stimulate the production and accumulation of LDs in HCV-infected cell cultures [74]. During HCV replication, the core protein is mobilized from the ER to the LDs after its synthesis for temporary storage before being transported back to the ER to assemble new viral particles [75]. Since the core protein has also been shown to affect lipid formation and accumulation, a relationship between lipid metabolism and regulation of viral production is suggested [74].

Another condition that can be caused by HCV infection is non-alcoholic fatty liver disease (NAFLD), which is also diagnosed by an exacerbation of LD. It is estimated that this condition affects up to one-third of the world’s population [76,77]. This disease is associated with dysregulation of oxidative pathways and lipid synthesis; in this sense, it has been found that the process of de novo lipid synthesis plays a vital role during infection. In addition, an increase in proteins that regulate lipid biogenesis, such as the transcription factor sterol regulatory element binding protein (SREBP1) and carbohydrate responsive element binding protein, has been observed in patients with this disease and in murine models [78].

In contrast, in these same models in humans and murine, an imbalance in mitochondrial fatty acid oxidation has been observed, which triggers hepatic steatosis [78,79]. As mentioned previously, there is a family of proteins surrounding LD known as Perilipins or PAT family, which in addition to serving as protection of LD, are also involved in developing diseases caused by HCV [80,81,82]. In a mouse model of NAFLD, it was observed that perilipin 2 and 3 proteins could protect the liver from lipotoxicity caused by NAFLD [82]. However, perilipin 1 and 2 proteins are upregulated in humans with NAFLD. Furthermore, Perilipin 2 is an excellent human marker for LD and liver disease [83,84].

### 3.2. Flavivirus

Flaviviruses, including Zika virus (ZIKV), dengue virus (DENV), and West Nile virus (WNV), interact with LDs through their protein components. ZIKV, which spread widely during an outbreak in Brazil between 2014 and 2015, has generated significant research, not only due to its pathogenesis but also due to the medical implications associated with newborns, who may experience complications during and after delivery, such as microcephaly and neurological complications [85]. Furthermore, it is estimated that close to 500 million people in the Americas are at risk of contracting DENV. These data are particularly relevant given that DENV and ZIKV coexist in the same geographic regions, which has created challenges in diagnosing both infections [86]. WNV can cause a fatal nervous system disease and is commonly found in Africa, Europe, the Middle East, North America, and western Asia. These flaviviruses (DENV, ZIKV, and WNV) are transmitted by *Aedes* sp. (mosquitoes) distributed in the tropics [87].

In this context, LDs play a crucial role in the pathogenesis of flaviviruses since these viruses appropriate LD functions and use them for their assembly and replication. It has been shown that an alteration in lipid metabolism occurs during flavivirus infections [88]. An increase in LDs has been observed in the early stages of infection, suggesting that these viruses promote LDs biogenesis to facilitate their replication and assembly. However, in the later stages of infection, the amount of LDs decreases, which is attributed to their selective autophagy [89,90,91].

Both the biogenesis and the degradation of LDs have been demonstrated in in vivo and in vitro systems during infections by these viruses. In addition, the helix–loop–helix structure of the capsid protein has been shown to interact with LDs. It has also been shown that the inflammatory and antiviral responses caused such infection are closely related to LDs [89,92,93,94].

DENV induces proviral selective autophagy (lipophagy) directed at LDs, which stimulates lipid metabolism by increasing β-oxidation associated with increasing cellular ATP. In contrast, exogenous lipids can complement autophagy for further DENV replication. Zhang et al. demonstrated that NS4A/B proteins bind to unmodified lipid droplet regulating VLDL assembly factor (AUP1) and promote its translocation from LDs to autophagosomes to drive lipophagy induction [92,95].

ZIKV infections reveal the significant importance of prostaglandins in the course of infection. Studies of lipid profiles in infected patients and *Aedes* sp. have shown increased prostaglandin H2. This increase can trigger the formation of other prostaglandins, thus amplifying their impact. In addition, an increase in prostaglandin precursors such as ethanolamine phosphate and arachidonic acid, which regulate the production of various prostaglandins, including prostaglandins E2 and H2, was observed [96].

Interestingly, metabolomic studies have shown that ZIKV uses LDs for prostaglandin biogenesis in the human placenta. The crucial implication here is that inflammation of this tissue by prostaglandins could be connected to the development of microcephaly or damage to the placenta in newborns. This is a potentially critical aspect of ZIKV pathogenesis, highlighting the importance of LDs and prostaglandins in viral infections [97].

### 3.3. Respiratory Syncytial Virus (RSV)

Studies have shown that during RSV infection, certain prostaglandins can be detected. In the context of such infection, prostaglandin E2 (PGE2) interaction has been observed to occur independently of viral or cellular protein synthesis. RSV infection induces the release of arachidonic acid through the induction of cytoplasmic phospholipase A2 (cPLA2) enzymatic activity and its translocation into the membrane. Specific cPLA2 inhibitors significantly block RSV-induced PGE2 secretion, indicating a pivotal role for cPLA2 in virus-induced PG production. Blockade of PG secretion, through inhibition of cPLA2 or COX-2, results in impaired RSV replication and subsequent RSV-mediated epithelial cell responses, suggesting that inhibition of PG secretion could be beneficial in RSV infection by reducing the production of proinflammatory mediators [98,99,100].

In animal models, RSV promoted the dispersal and utilization of LDs, causing enlarged oxidative lesions and increased proinflammatory cytokines (IL-1, IL-2, IL-4, and IL-6), leading to the progression of airway hyperresponsiveness [101].

### 3.4. Influenza Virus

Influenza A virus, which causes an estimated 250,000 to 500,000 deaths annually worldwide, is a viral agent of great medical importance that interacts with LDs [102]. During influenza infection, this virus can induce changes in LDs and modulate lipid synthesis in host cells. In contrast to other viruses, influenza infection increases autophagy pathways. Specifically, increased activity of the mammalian target of rapamycin complex (mTORC)-2 or via mTORC1 is observed, which induces LD degradation during infection [103].

The increase in selective autophagy during infection, along with the interaction of lysosomes with LDs and the autophagolysosome formation, are strategies the virus uses for replication [104]. In addition, during influenza virus infection, the production of proinflammatory cytokines is increased, including interferon-α (IFN-α), IFN-β, tumor necrosis factor-α (TNF-α), interleukin-6 (IL-6), and monocyte chemotactic protein 1 (MCP-1). Furthermore, it has been observed that the influenza virus interferes with the lipid metabolism of the host cell, manipulating proteins involved in lipid regulation, such as the interferon-induced viperin (IFN) protein, lipoxygenase 12/15, lipid metabolite ω-3, and protectin D1 (PD1), which is derived from polyunsaturated fatty acids. The influenza virus downregulates both viperin and PD1, which facilitates infection [105,106].

Viperin, which colocalizes with cytosolic LDs, acts as an efficient antiviral agent against various viruses, including hepatitis C, dengue, human cytomegalovirus, West Nile virus, and influenza virus. This protein exerts its antiviral function by modulating lipid rafts and plasma membrane regions rich in glycosphingolipids and cholesterol [107]. Given this composition, the plasma membrane in these areas is less fluid than in the rest. Specifically, LDs supply neutral lipids and cholesterol to these lipid rafts. Thus, these rigid domains in the cell membrane provide a platform for several receptors involved in cell signaling. They also mediate the internalization of pathogens through different mechanisms of entry and immune response and are used by influenza virus for cell lysis [108].

Given this, viperin inactivates the enzyme farnesyl pyrophosphate synthase (FPPS), which drives the formation of lipid rafts. Without these structures, newly formed influenza virions cannot be released, thus preventing infection of new host cells [109].

In contrast, the PD1 protein, which is also downregulated by the influenza virus, has a relevant role in the pathogenesis of the infection. PD1 suppression leads to activation of lipoxygenase 12/15. This enzyme has a fundamental role in the immune response and the synthesis of proinflammatory cytokines such as IL-1β and the formation of ROS by macrophages, and its activation has been associated with increased lethality and pathology of influenza infection. Therefore, manipulating lipid metabolism by the influenza virus, including the downregulation of PD1 and viperin, is an essential mechanism by which the virus promotes its spread and pathogenicity [105,110]. Due to the essential role of lipids in the influenza virus replication cycle, lipid-lowering drugs such as atorvastatin have been explored, inhibiting the formation of LDs and their use in the virus replicative cycle [111].

### 3.5. SARS-CoV-2

The current pandemic, caused by the respiratory virus SARS-CoV-2, has generated an unprecedented effort in the scientific community to elucidate the mechanisms of virus entry and host–pathogen interactions. It is essential to highlight that LDs have been shown to play a crucial role in the development of infection by this virus [88,112].

Recently, a team of researchers showed that SARS-CoV-2 infection induces the activation of proteins involved in lipogenesis, including PPARγ and SREBP-1, and promotes LD biogenesis in monocytes from infected patients. This finding highlights the critical importance of LDs during SARS-CoV-2 infection [112,113].

In a cellular model, tests were carried out in which the proteins that participate in the formation of LDs were inhibited using a drug (xanthohumol) that blocks the function of the DGAT protein, which is essential for the formation of LDs by converting diacylglycerols into triacylglycerols, constituents primordial of the LDs. Notably, this pharmacological inhibition of LDs decreased viral replication [114]. In another study to quantify cell damage caused by a viral infection, the release of lactate dehydrogenase, an indicator of cell death, was measured. The results showed increased cell death in response to viral infection, and infected cells exhibited morphological changes consistent with necrosis. Furthermore, the data collected by this research team suggest that SARS-CoV-2 uses LDs to replicate and assemble new viral particles and that pharmacological targeting toward LD formation may inhibit SARS-CoV-2 replication. This finding opens a potential avenue for developing antiviral strategies against SARS-CoV-2 [114].

LDs are organelles with essential roles in producing inflammatory mediators and innate signaling in immune cells [38]. SARS-CoV-2 infection induces a dysregulation of the proinflammatory immune response, characterized by the activation of various cytokines and chemokines. Particularly noteworthy is the activation of leukotriene LTB4, a bioactive molecule that can potentially induce the production of ROS [102,103]. Simultaneously, the CysLTR1 receptor emerges as a critical player in the context of this infection. Its activation leads to bronchoconstriction, a phenomenon that significantly contributes to inflammation of the pulmonary adipose tissue, a common finding in viral infections. Additionally, it has been identified that the activation of the CXCL10 chemokine can be a reliable marker of cardiac and pulmonary damage, expanding the understanding of the systemic effects of SARS-CoV-2 infection [115]. Regarding the influence of the virus on other inflammatory mediators, an increase in the production of interleukin-12 (IL-12) has been documented in contrast to a decrease in the expression of interleukin-4 (IL-4) when compared to infected monocytic cells with uninfected cells. This disturbance in the balance of cytokines suggests a modulation of the immune response during SARS-CoV-2 infection [116,117].

These alterations in the immune response may have significant implications for the clinical course of COVID-19. Activation of the lipid-derived molecule leukotriene LTB4 and the consequent production of ROS may contribute to the severity of the inflammatory condition, exacerbating tissue damage and potentially leading to complications such as acute respiratory distress syndrome (ARDS) [118,119]. Furthermore, bronchoconstriction induced by CysLTR1 activation can impair lung function, leading to decreased respiratory capacity and the need for respiratory support [120,121]. Regarding the cytokine imbalance, the increased production of IL-12 and decreased IL-4 suggest that SARS-CoV-2 infection may promote a Th1 immune response associated with inflammation and effector cell activation such as macrophages. These cells may be essential for forming LDs, as activated macrophages have been shown to create LDs in their response to infections [1,122].

Finally, the proinflammatory modulation during SARS-CoV-2 infection can be suppressed by pharmacological inhibition of the DGAT-1 enzyme using the drug A922500. This intervention strategy disrupts the formation of LDs, a consequence that, in turn, has been shown to reduce the production of proinflammatory response mediators markedly. Additionally, the viral load is reduced. This fact underscores the importance of LDs in synthesizing these inflammation-promoting mediators. This suggests that LDs could be essential for the immune response and disease progression in the context of SARS-CoV-2 infection, providing a potential therapeutic target to mitigate the inflammatory response associated with this viral disease [114].

### 3.6. Rotavirus

Recently, it has been shown that there is an association between LDs and rotavirus (RV) infection, a pathogen that mainly affects the small intestine’s villi and is the leading cause of gastroenteritis in infants and children worldwide. This virus is hazardous, as it can cause vomiting and diarrhea, which, if not treated properly, can lead to dehydration and be life-threatening [123].

In a 2022 study, Criglar et al. demonstrated the interaction between the NSP2 protein of RV and the PLIN-1 protein of LDs. This interaction favors the formation of viroplasms in infected cells, which are essential for the assembly of viral particles and the biogenesis of LDs [124]. Viroplasms are electronically dense structures in the cytoplasm of infected cells and are composed of viral and cellular proteins [125]. In the case of RV, several of its proteins, both structural and non-structural, as well as components of the LDs, are detected in viroplasms. These spherical structures have a core of neutral lipids and are decorated with various cellular proteins, including those associated with LDs, such as perilipins (PLIN 1-5) [124].

However, the exact relationship between the rotavirus NSP2 protein and LDs remains unclear. The NSP2 protein exists in two structurally distinct forms: dispersed in the cytoplasm and localized in viroplasms. However, whether NSP2 or NSP5 physically interact with LD components to trigger their biogenesis is still unknown. Clinically, the alteration of LDs caused by RV infection can trigger gastroenteritis that can be life-threatening in some cases if not treated immediately, compared to other enteric diseases [124].

As mentioned above, cancer cells and animal models have been used to study the relationship between rotavirus infection and lipid droplets. However, recently the enteroid model has been used, which has cultures that resemble the intestinal wall and are an excellent model for research. In this case, enteroids have been cultured and infected with rotavirus, and it was observed that the infection by this virus induces an exacerbation of lipid droplets. It was confirmed that the virus takes advantage of the lipogenesis pathways and lipid droplets as structural platforms for virus replication [126].

### 3.7. Adenovirus

Adenovirus type 2 (AD-2) infection has been shown to cause inflammation in the alveolar cells of the upper respiratory tract. This phenomenon is attributed to an increased accumulation of cholesterol esters in lipid droplets (LD). This accumulation activates a protein complex known as oxysterol-binding protein 1L (ORP1L), which regulates the activation of the late endosome of the cells. In adenovirus type 5 (AD-5) infection, another virus affecting the respiratory tract, a similar accumulation of neutral lipids has been observed. This has been correlated with elevated levels of total cholesterol, low-density lipoprotein (LDL), and high-density lipoprotein (HDL). This was observed when analyzing serum from AD-5 seropositive individuals using an automated biochemical analyzer and flow cytometry; however, triacylglycerol levels in monocytes derived from AD-5 seropositive individuals were lower compared to control monocytes from AD-5 seronegative individuals [27,126]. This increase in lipoproteins led to an increase in reactive oxygen species (ROS) in addition to activating the innate immune response observed by elevated levels of interleukins and interferons which activated potent antiviral inflammatory responses. Some studies mention that the immunometabolism which includes changes in lipid metabolism may possibly be associated with the development of cancer; however, it requires further study to determine if the changes produced by infections by this virus and lipid droplets can trigger oncogenic events [126].

### 3.8. Rabies Virus

Rabies, a zoonotic disease caused by the rabies virus (RABV), is characterized by its neurological effects and destructive potential without adequate post-exposure treatment [127]. Although it is practically eradicated in many places, a close relationship between lipids and the rabies virus has been observed.

The NDRG1-DGAT1/2 pathway has been found to play a critical role in the exacerbation of lipid droplets during rabies virus infection. In addition, pharmacological inhibition of lipid droplet formation by inhibiting the enzyme HMG-CoA reductase with atorvastatin has been found to reduce lipid droplet formation and decrease viral particle production. It should be noted that atorvastatin, besides lowering cholesterol levels, may have multiple additional effects that could influence the immune response and the pathogenesis of rabies disease. Additionally, it has been found that the viral proteins RABV-M and RABV-G, mainly involved in the viral-budding process, can colocalize with LDs, suggesting that the rabies virus can use the LDs as a vehicle to facilitate the release of viral particles and ultimately increase viral production. This report highlights the relevant role of lipid droplets in RABV replication. It points out that its biogenesis is regulated through the NDRG1-DGAT1/2 pathway, providing new opportunities for drug development directed against RABV [128].

As mentioned above, the disease produced by the rabies virus is zoonotic, which means that it can be transmitted from animals to humans. In this sense, the studies in which a deregulation of lipid droplets has been observed that infection by this virus causes an up-regulation of NDRG1 and DGAT proteins and these in turn promote an exacerbation of lipid droplets could be associated with lipotoxicity events in the animal. However, it remains to be understood what other pathological events can be triggered by studying the NDRG1-DGAT-Lipid droplets complex associated with infections by this virus.

### 3.9. Cytomegalovirus

Human cytomegalovirus (HCMV) is a genus of herpesviruses within the subfamily Betaherpesvirinae. One of the main functions of human cytomegalovirus is the modulation of cellular metabolism to promote replication processes and allow a successful cycle. It has been shown that infection by this virus in adipocyte cells can cause AMPK activation and this in turn can activate glucose transporter 4 (GLUT4). The result of this process is the increase in the biogenesis of lipid droplets through the activation of the transcription factor chREBP, which favors the virus assembly events. In contrast, it has been observed that the accumulation of lipids caused by infection of this virus can decrease the levels of ATP available in the cells and affect the cytoskeletal actin of the infected cells. These events were demonstrated when the expression of the viperin protein of the virus was inhibited [129]. However, recent studies have shown that cytomegalovirus may be a viral promoter of oncogenesis since some of its proteins have been found in tumors. Interestingly, an increased amount of lipid droplets has been found in latent polyploid giant cancer cells (PGCC) from breast cancer biopsies in cytomegalovirus-infected patients [130].

## 4. Conclusions

In conclusion, the investigation of LDs has unveiled their multifaceted nature in viral interactions and the development of human diseases. These intracellular structures play a pivotal role in the replication, assembly, and release of diverse viruses, making them promising pharmacological targets for inhibiting viral infections. This review has explored how different viruses, such as HCV, DENV, ZIKV, WNV, RSV, influenza virus, RV, AD, RABV, and HCMV interact with LDs to promote their replication and spread. In many cases, inhibiting LDs formation or manipulating their lipid content has effectively suppressed viral replication and reduced viral load. Furthermore, the significance of LDs in immune response and the pathogenesis of viral diseases has been highlighted. Dysregulation of cytokines, chemokines, and inflammatory mediators associated with LDs can lead to imbalanced immune reactions and exacerbation of inflammation in various viral infections. Given these findings, LDs emerge as a promising therapeutic target for developing antiviral drugs. Inhibiting LD formation, manipulating lipid content, or interfering with crucial protein interactions within LDs can offer novel therapeutic strategies to combat viral infections. However, it is essential to acknowledge that targeting LDs pharmacologically presents challenges and limitations. A deeper understanding of the underlying molecular and cellular mechanisms is needed to optimize the efficacy and safety of LDs-targeted therapies. In summary, the study of lipid droplets has unveiled their central role in viral interactions and the development of human diseases. Manipulating LDs provides a promising approach to inhibiting viral replication and may serve as a valuable strategy in the battle against various viral infections. Continued research and exploration of LDs-targeted therapies are crucial to enhance the ability to combat viral diseases and improve human health.

## Figures and Tables

**Figure 1 microorganisms-11-01851-f001:**
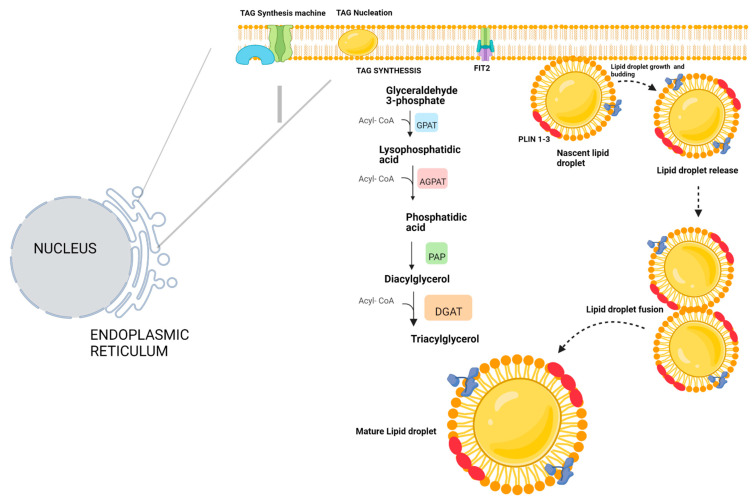
Biogenesis of lipid droplets. Lipid droplets originate in the endoplasmic reticulum. Triacylglyceride-producing enzymes are in the endoplasmic reticulum membrane. Triacylglycerides accumulate between the two phospholipid monolayers, and these structures grow into a nascent lipid droplet, which eventually grows large enough to separate from the endoplasmic reticulum. Additionally, during the process, LDs are coated by a series of proteins known as perilipins, which, although they are not the only proteins that can be incorporated into LDs, are those that are found in greater abundance in LDs. Once the LDs are formed with the triacylglyceride nucleus and sterol esters and coated by proteins, they separate from the endoplasmic reticulum, where they can fuse with other LDs to form mature LDs.

**Figure 2 microorganisms-11-01851-f002:**
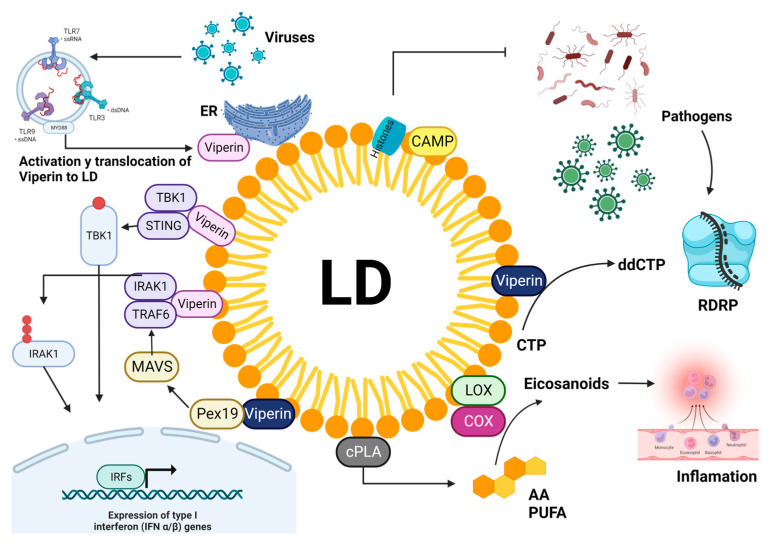
Involvement of LDs in the immune system. One of the proteins that can interact with LDs after a viral infection is viperin. This protein is synthesized from signaling through TLRs. It is one of the main proteins belonging to ISG, which alone can have an antiviral effect as it can catalyze CTP to ddCTP, a ribonucleotide that inhibits the synthesis of viral RNA. However, for there to be a more potent and antiviral effect, ISGs usually bind to more proteins with the same purpose; in this sense, it has been observed that STING can bind to LDs after that Viperin has done so, and form a more robust complex with more significant antiviral activity. Another of the bonds observed when Viperin acts in LDs is the participation of Pex19, which is fundamental for forming peroxisomes that favor the antiviral response. Cathelicidins which are small molecules shown to have antiviral activity, can also be associated with LDs. The enzymes lipoxygenase and cyclooxygenase involved in activating eicosanoids from polyunsaturated fatty acids may also be associated with LDs and, during viral infections, may trigger inflammatory events.

**Figure 3 microorganisms-11-01851-f003:**
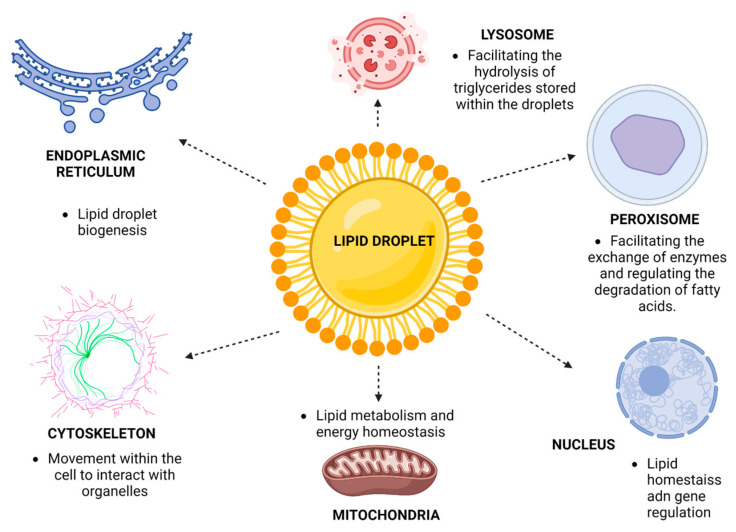
Lipid droplet organelle interactions. The biogenesis of LDs begins in the ER; however, once mature, LDs can re-associate with the ER to recycle LD components, such as lipids and proteins, for the LDs to remain functional. LDs use the tubulin networks of the cytoskeleton to mobilize within the cell and interact with other cellular organelles. In the case of mitochondria, the LDs-mitochondria interaction occurs so that free fatty acids can be delivered so that beta-oxidation can take place and energy can be obtained in the form of ATP. In the nucleus, it is known that there are nucellar LDs (nLDs), which participate in the release and metabolism of phospholipids that serve for the remodeling of the chromatid. It has also been seen that the nLDS can impact gene regulation because they can remodel components associated with the genetic material. In the case of peroxisomes, there is evidence that both organelles are formed in the same area of the ER. Both organelles could interact to carry out the degradation of LDs and to have free fatty acids available that can be used for beta-oxidation. Additionally, this interaction helps recycle the protein components of the LDs to remain functional. The relationship between LDs and lysosomes is due to a process known as lipophagy, which is simply the degradation of LDs to obtain free fatty acids, which can be used to obtain energy or to maintain lipid membranes.

**Figure 4 microorganisms-11-01851-f004:**
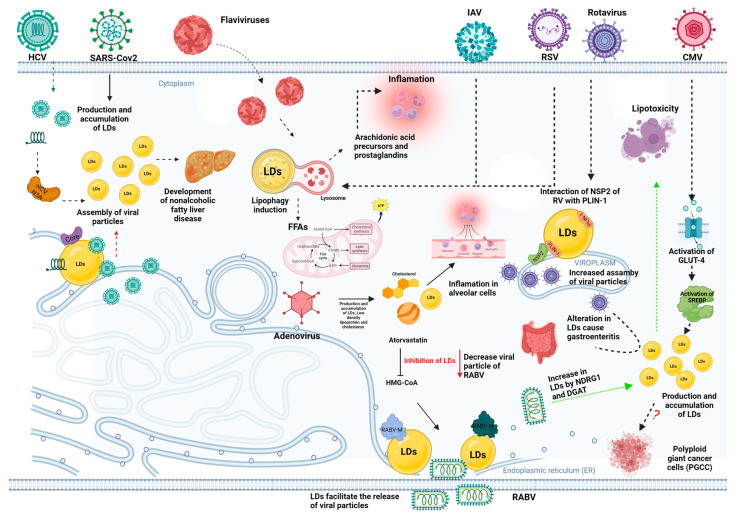
The role of LDs during viral infection. During HCV infection, there is an increase and accumulation of LDs; this increase is related to the appearance of non-alcoholic fatty liver. In the case of flaviviruses, LDs are degraded through lipophagy to obtain FFAs. However, the formation of FFAs can favor the formation of arachidonic acid and precursors of inflammatory molecules. SARS-Cov2 infection induces the activation of proteins involved in lipogenesis, including PPARγ and SREBP-1, and promotes LDs biogenesis to carry out the virus’s replication cycle. LDs are degraded during infection with influenza A virus (IAV) and respiratory syncytial virus (RSV). On the other hand, during adenovirus infection, it has been seen that there is an accumulation and production of LDs related to the formation of an inflammatory state within the target cells. In the case of rabies virus (RABV), there is an increase in LDs mediated by the enzymes DGAT and NDRG1 (green arrow), which facilitate the release of viral particles. However, this increase also generates a lipotoxic state within the cells. In addition, it is known that inhibition of LDs through the drug Atorvastatin affects the viral replication cycle (red arrow). In the case of Rotaviruses, the NSP2 protein can interact with the PLI-1 protein to increase the assembly of viral particles and the number of LDs, which is related to the appearance of gastroenteritis. Finally, in the case of cytomegalovirus, it has been observed that the increase in GLUT-4 receptors is associated with the activation of the enzyme SREBP, which in turn promotes the production and accumulation of LDs, which has been linked to the appearance of Polyploid cancer.

## Data Availability

Not applicable.

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
