# Peer review of "Multifaceted Nature of Lipid Droplets in Viral Interactions and Pathogenesis"

_microorganisms, 2023, doi:10.3390/microorganisms11071851_

Round 1

Reviewer 1 Report

The authors have summarized the lipid droplet (LD) biogenesis, LD interaction with other organelles, and the roles of LD in the virus replication and pathogenesis. Overall, the review is well-organized and of interest to readers in the field. However, concerns and comments are noted below. 

1. It is unclear whether LDs positively or negatively regulate inflammation during SARS-CoV-2 replication and pathogenesis.

 2. Line 364 introduces lipid rafts rich in glycosphingolipids and cholesterol. It is recommended that the authors add a brief explanation about the relationship between LDs and lipid rafts.

3. It is suggested that the manuscript is proofread and edited as needed, as some sentences are not clearly written or have errors. For example, line 236, “Depending on the organelle with which they interact, it will be the mechanism that can regulate.”; line 514, “Figure 3. The paper of LDs during virus infection.”; etc.

4. It is suggested that the authors make more efforts to improve the Figure 3. Figure 3’s legend needs to be more detailed to explain most of the processes depicted in the figure. On the bottom of the figure 3, does RAVB indicate RABV? SARS-CoV-2 is not shown in the figure 3?

Minor editing is recommended.

Author Response

Dear Reviewers

I am pleased to resubmit for publication the revised version of “Multifaceted Nature of Lipid Droplets in Viral Interactions and Pathogenesis”.  I appreciated the constructive criticism from the associated editor and reviewers. I have addressed each of their concerns as outlined below.      

Following the reviewer’s advice, I, along with my collaborators have been carefully revised and appropriate changes have been made in accordance with the reviewer’s suggestions. The responses to their comments are provided below:

We appreciate the recommendation and have contacted a service for correction of style and writing in English, through which the manuscript has been polished. In addition, a certificate of the company we hired will be attached.

Reviewer 1:

1.- 1. It is unclear whether LDs positively or negatively regulate inflammation during SARS-CoV-2 replication and pathogenesis.

We consider relevant comment to reviewer, and basically, we have added a paragraph where we describe this situation more clearly. These changes can be reviewed in the attached file, highlighted in yellow. LDs positively regulate inflammation during SARS-CoV-2 infection, and inhibition of LD formation reduces proinflammatory mediators and viral load. This was added in lines 553-561.

  1. Line 364 introduces lipid rafts rich in glycosphingolipids and cholesterol. It is recommended that the authors add a brief explanation about the relationship between LDs and lipid rafts.

We appreciate the observation, we corroborated that in the manuscript a sentence and   brief explanation of the relationship between LDs and lipid rafts was added in the lines 480-491.

  1. It is suggested that the manuscript is proofread and edited as needed, as some sentences are not clearly written or have errors. For example, line 236, “Depending on the organelle with which they interact, it will be the mechanism that can regulate.”; line 514, “Figure 3. The paper of LDs during virus infection.”; etc.

Thank you for your comments. The manuscript was sent to a grammar editing service to resolve these errors. We attach proof of said service.

  1. It is suggested that the authors make more efforts to improve the Figure 3. Figure 3’s legend needs to be more detailed to explain most of the processes depicted in the figure. On the bottom of the figure 3, does RAVB indicate RABV? SARS-CoV-2 is not shown in the figure 3?

We appreciate your comment. Have made the changes and corrections related to this comment. The figure legend mentioned more details in relation with the figure and the text errors have already been corrected.

Reviewer 2 Report

The review entitled “Multifaceted Nature of Lipid Droplets in Viral Interactions and Pathogenesis” authored by Huitron et al reviews the various roles that lipid droplets play in viral infections. This review is in general, comprehensive and well written however I have outlined the below points that would enhance the impact of this review.

Major Comments:

-          There are several studies missing that should be discussed in regards to lipid droplets forming contacts with other organelles. Lipid droplets can form contacts with other organelles for a plethora of reasons, however, it is important for this review to discuss the antiviral roles, or the immune specific roles following virus infection with the authors fail to do. For example, lipid droplets have been found to be associated with the ER (and STING) to form innate immune signalling complexes with other antiviral pathway members such as viperin; https://doi.org/10.1111/imcb.12420 . There is also some evidence of lipid droplets associating with peroxisomes to form signalling complexes during viral infection. This would be a good addition to the section discussing lipid droplet associations with other organelles as it demonstrates a direct mechanism for innate immune signalling augmentation. See: https://www.life-science-alliance.org/content/4/7/e202000915

-          Lipid droplets have been shown to be upregulated in flavivirus infections in both mammalian cells and mosquito cells, however this is not discussed in this review. This upregulation has now been demonstrated to be essential for innate immune signalling activity so is important to be included in this review. Please see: https://www.nature.com/articles/s41467-021-24632-5, https://www.mdpi.com/1422-0067/22/9/4418, https://www.nature.com/articles/srep19928 (and others). The addition of a section discussion this is essential to this review as it underpins the authors main arguments in the abstract and the title. 

The quality of the English language in this manuscript is great. Only minor editing would be needed.

Author Response

We appreciate and consider the reviewer's observations and suggestions highly relevant. We placed a new section on the role of LDs and the immune response. In this, we mention the most relevant evidence in this context. In addition to adding an image about this topic, these changes can be observed in the new version marked in yellow can be identified.

Finally, we again thank you for your suggestions and insights, which have enriched the manuscript and produced a more balanced and better account of the review. We hope that the revised manuscript is now suitable for publication in the prestigious journal that you represent.

Round 2

Reviewer 2 Report

The authors have addressed all of the comments that I suggested, therefore, I think the manuscript has been improved and would be suitable for publication. 

Only minor English language edits need to be made